# Methylation of recombinant mononucleosomes by DNMT3A demonstrates efficient linker DNA methylation and a role of H3K36me3

Alexander Bröhm[1], Tabea Schoch[1], Michael Dukatz[1], Nora Graf[1], Franziska Dorscht[1], Evelin Mantai[1], Sabrina Adam[1], Pavel Bashtrykov[1] & Albert Jeltsch [1✉]

Recently, the structure of the DNMT3A2/3B3 heterotetramer complex bound to a mononucleosome was reported. Here, we investigate DNA methylation of recombinant unmodified, H3K$_C$4me3 and H3K$_C$36me3 containing mononucleosomes by DNMT3A2, DNMT3A catalytic domain (DNMT3AC) and the DNMT3AC/3B3C complex. We show strong protection of the nucleosomal bound DNA against methylation, but efficient linker-DNA methylation next to the nucleosome core. High and low methylation levels of two specific CpG sites next to the nucleosome core agree well with details of the DNMT3A2/3B3-nucleosome structure. Linker DNA methylation next to the nucleosome is increased in the absence of H3K4me3, likely caused by binding of the H3-tail to the ADD domain leading to relief of autoinhibition. Our data demonstrate a strong stimulatory effect of H3K36me3 on linker DNA methylation, which is independent of the DNMT3A-PWWP domain. This observation reveals a direct functional role of H3K36me3 on the stimulation of DNA methylation, which could be explained by hindering the interaction of the H3-tail and the linker DNA. We propose an evolutionary model in which the direct stimulatory effect of H3K36me3 on DNA methylation preceded its signaling function, which could explain the evolutionary origin of the widely distributed "active gene body-H3K36me3-DNA methylation" connection.

[1] Department of Biochemistry, Institute of Biochemistry and Technical Biochemistry, University of Stuttgart, 70569 Stuttgart, Germany.
✉email: albert.jeltsch@ibtb.uni-stuttgart.de

DNA methylation plays essential roles in gene regulation and chromatin biology[1–3]. The DNMT3A and DNMT3B paralogs are de novo DNA methyltransferases[4,5], which generate DNA methylation during gametogenesis and post-implantation development[6,7]. Additionally, the catalytically inactive homolog DNA methyltransferase 3-like protein (DNMT3L) acts as a regulatory and stimulating factor for DNMT3A in germ cells and is required for the establishment of maternal imprints and sperm development[4]. Structural analyses showed that the C-terminal domain of DNMT3L (DNMT3LC) associates with the catalytic domain of DNMT3A (DNMT3AC) or DNMT3B (DNMT3BC) to form linear hetero-tetramers with DNMT3LC at the edges and two subunits of DNMT3AC or DNMT3BC in the center[8–10]. In the following, C-terminal domain fragments of DNMT3 proteins will be indicated by adding a C to the name, complexes will be abbreviated as DNMT3AC/3LC, as appropriate depending on their composition. Biochemical studies demonstrated that in the absence of DNMT3L, additional DNMT3A subunits can take the places of DNMT3L, leading to the formation of DNMT3AC and DNMT3A2 tetramers and higher order complexes[11–14]. Methylation of DNA by DNMT3A in a nucleosomal context is preferentially occurring on the linker DNA and is inhibited by the presence of the linker histone H1[15–17]. Therefore, nucleosome positioning acts as an important regulatory mechanism for DNMTs[15,18,19]. Furthermore, the activity of DNMTs is dependent on their distinct specificity for flanking DNA sequences[10,20–22]. The activity of DNMT3 enzymes is further connected to histone modifications by the chromatin interaction of their PWWP and ADD domains[4,5]. The PWWP domain is a reader of di- and trimethylation of lysine 36 on histone H3 (H3K36me2/me3) and directs DNMT3A to pericentromeric heterochromatin[23–26]. Moreover, H3K36 methylation-dependent targeting of DNMT3B to gene bodies and DNMT3A to intergenic regions has been observed[27,28]. Disruption of the K36me2/3 binding pocket in the PWWP domain of DNMT3A leads to genome-wide aberrations in DNA methylation patterns and diseases[29–31]. The ADD domain specifically binds the N-terminal part of the histone H3-tail, if lysine 4 is unmodified and it also acts as an allosteric activator of DNMT3A and DNMT3B[16,32–35].

In a recent seminal paper, the cryo-EM structure of DNMT3A bound to a mononucleosome was reported[36]. In this study, DNMT3A2 was complexed to DNMT3B3, a catalytically inactive splicing isoform of DNMT3B that is expressed in differentiated cells and tumors and stimulates DNMT3A in a manner reminiscent of DNMT3L[36,37]. In the cryo-EM structure, the DNMT3A2/3B3 complex was observed to form a similar linear heterotetramer as the DNMT3A/3L catalytic domains, in which DNMT3B3 replaces DNMT3L at the outer complex positions. Unexpectedly, one of the DNMT3B3 subunits was found to bind with the residues R740 and R743 to the H2A/H2B acidic patch on the disc face of the histone octamer, thereby anchoring the DNMT3A2/3B3 tetramer on the nucleosome core particle and providing an orientation towards the linker DNA strand near the dyad axis. The amino acid sequences of DNMT3A and DNMT3B are highly similar at the contact point to the nucleosome acidic patch, suggesting that a DNMT3A2 homotetramer could form a similar interaction (Supplemental Fig. 1). In agreement with structures of DNMT3AC/3LC with naked DNA[9], DNA interaction of the complex is limited to the central tetramer interface comprising the two DNTM3A2 subunits, which bind the linker DNA. Considering the conformational changes that accompany base flipping, the distances between the DNA bases and the active site pockets of the DNMT3A2 subunits in the nucleosomal complex suggest that the distal DNMT3A subunit could access cytosine residues placed in the linker DNA region. In contrast, the proximal DNMT3A subunit is unlikely to bind a flipped cytosine, because of its positioning next to the nucleosome core and the large distance to the DNA. In this arrangement, it would be plausible to assume that the distal DNMT3A is able to methylate a CpG-site located close to its catalytic site, while the nucleosome-proximal subunit should not show catalytic activity. However, detailed and quantitative data regarding the methylation activity of DNMT3A2 on mononucleosomes at base pair-resolution are not available so far. Moreover, the mononucleosomes used in the cryo-EM structure did not contain H3K36me3 and the DNMT3A2 ADD and PWWP domains as well as the H3 N-terminal tails were not resolved. Hence, the individual roles of these elements in DNA methylation remained unclear.

In this study, we investigated DNA methylation of unmodified, as well as H3K4 and H3K36 trimethyllysine analog containing mononucleosomes by DNMT3A2, DNMT3AC, and the DNMT3AC/3B3C complex. We observed protection of nucleosome bound CpG sites against methylation, but efficient methylation of the nucleosomal linker DNA that reflects the flanking sequence preferences of DNMT3A on free DNA. In agreement with the cryo-EM structure, we observed that DNMT3A2 methylates the linker DNA at the position of the distal DNMT3A2 subunit, but not at a CpG site closer to the nucleosome. Competitive methylation experiments showed that H3K$_C$4me3 reduced methylation at CpG sites close to the nucleosome core, while H3K$_C$36me3 has a stimulatory effect on the methylation of the linker DNA in mononucleosomes. Strikingly, experiments with DNMT3AC lacking the PWWP and ADD domains revealed an even stronger stimulation of linker DNA methylation by H3K$_C$36me3. These data document a previously unrecognized role of H3K36me3 in directly stimulating linker DNA methylation by DNMT3A that is not dependent on its interaction with the PWWP domain of DNMT3A2. Fluorescence spectroscopy data suggest that K36 methylation hinders the interaction of the H3-tail with the linker DNA and thereby provides better access for DNMT3A to the DNA, which could explain its effect on DNA methylation. Based on this we propose an evolutionary model, in which the direct stimulation of DNA methylation by H3K36me3 preceded its PWWP domain-mediated signaling effect.

## Results

In this study, we aimed to investigate DNA methylation of unmodified as well as H3K$_C$4me3 and H3K$_C$36me3 containing mononucleosomes by DNMT3A2, DNMT3AC, and the DNMT3AC/3B3C heterotetramer at CpG site resolution using bisulfite-sequencing coupled to NGS. To generate a suitable methylation substrate, we designed a 240 bp long DNA substrate for nucleosome formation that is based on the Widom-601 nucleosome positioning sequence[38], but has an extended 5′ linker DNA length (Fig. 1a). The substrate contains 21 CpG sites, the first 7 of them in the 5′ linker DNA and the rest distributed throughout the nucleosome binding sequence. Both terminal DNA regions were designed to be free of CpG sites and used as primer binding sites during NGS library generation. An MluI restriction site was introduced in the middle of the 601 binding sequence to allow the cleavage of residual free DNA in the nucleosome preparations. To study the effect of the H3K4me3 and H3K36me3 modifications, the K4C or K36C mutations were introduced into the H3 expression construct after exchanging C96 and C110 to serine. Afterwards, the C4 or C36 residues in the purified H3 mutant proteins were converted into the trimethyllysine-analog using the alkylation reaction (Supplemental Fig. 2a, b)[39]. Histone octamers containing unmodified H3, H3K$_C$4me3, or H3K$_C$36me3 were assembled and purified by size

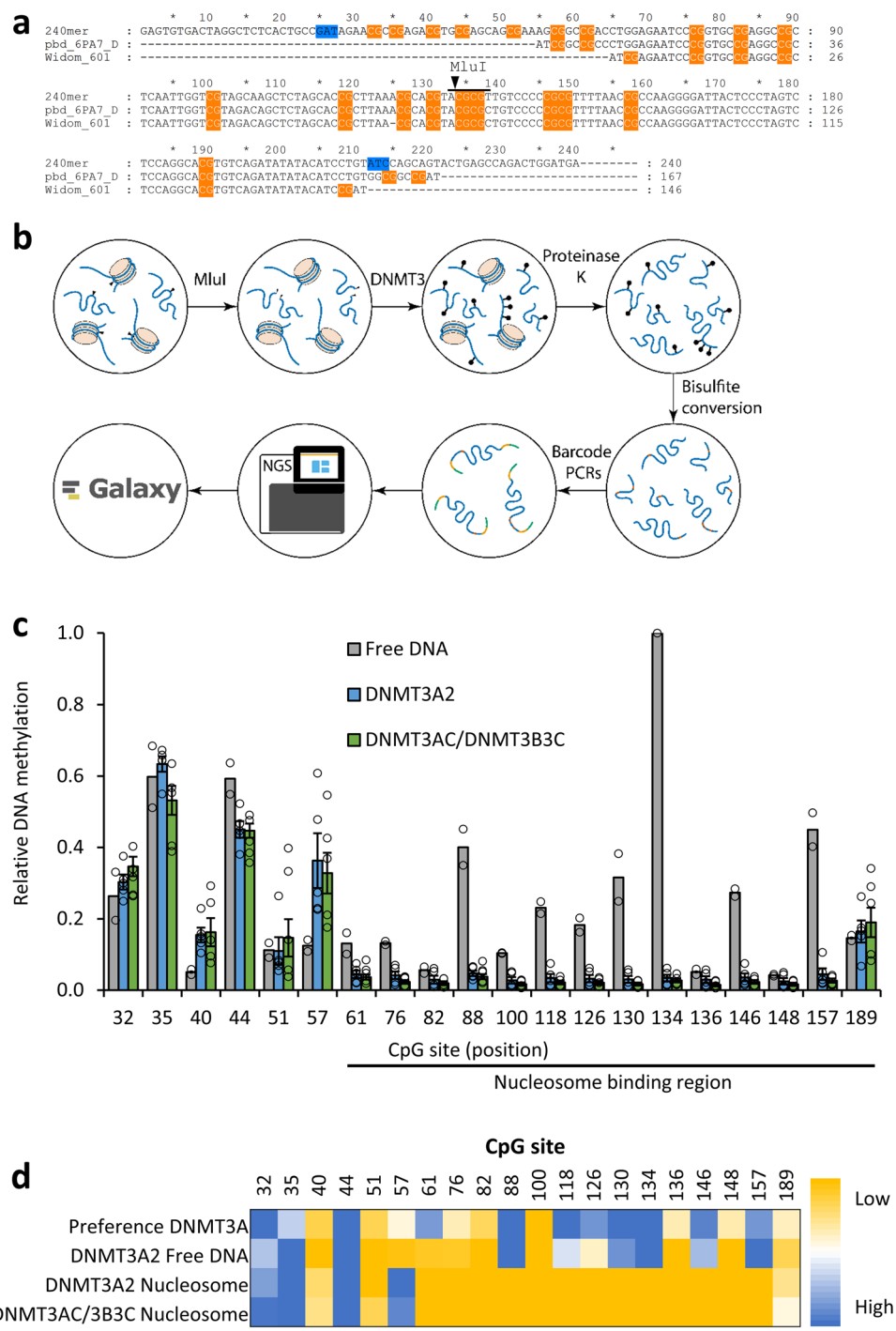

**Fig. 1 Methylation of nucleosomal DNA by DNMT3A2. a** Sequence alignment of the 167 bp nucleosome binding sequence used in the structure of the DNMT3A/3B3-nucleosome complex (pdb 6PA7), the original Widom-601 sequence[80], and the 240 bp sequence with extended CpG rich linker DNA used in our nucleosome reconstitutions. CpG sites are highlighted in orange and variable barcode sequences are shaded in blue. The MluI site is indicated. **b** Overview of the nucleosome methylation workflow. Unbound DNA is digested by MluI in the first step and the nucleosomes are then methylated by DNMT3A. All proteins are subsequently digested by proteinase K and the DNA is used for downstream processing consisting of bisulfite conversion, addition of barcodes, adapters, and indices by PCR, Illumina sequencing followed by bioinformatics analysis. **c** Relative methylation of each CpG site observed in five independent replicates of the nucleosome methylation with DNMT3A2 and DNMT3AC/3B3C, as well as two replicates of the free DNA methylation experiments. All nucleosome methylation experiments showed low methylation at CpGs in the nucleosome binding sequence on the nucleosomal substrate. For comparison, the relative intensities at the first 5 CpGs of the free DNA methylation were adjusted to the corresponding levels in the nucleosomal methylation. Error bars represent the SEM if $n \geq 3$. **d** Heatmap comparing the ±3 bp flanking sequence preference of DNMT3A[10] and the experimentally determined relative methylation levels on free DNA as well as on the nucleosomal substrate.

exclusion chromatography (Supplemental Fig. 2c, d). 240 bp DNA fragments with different internal sequence barcodes were prepared and used for reconstitution of unmodified, H3K$_C$4me3 and H3K$_C$36me3 nucleosomes by salt gradient dialysis. Successful nucleosome formation was confirmed by electrophoretic mobility shift assay (EMSA), where two prominent shifted DNA bands appeared in addition to the free DNA (Supplemental Fig. 3a, b). The phenomenon of two or more shifted bands after nucleosome reconstitution was described earlier[40] and can be explained by shifting of the octamer on the DNA most likely by one or two superhelical repeats. This results in an asymmetrically bound octamer that has a reduced electrophoretic mobility in comparison to a symmetric species.

**Investigation of nucleosomal DNA methylation.** The workflow for the nucleosome methylation experiments is shown schematically in Fig. 1b. First, residual unbound DNA from the reconstitution reaction was cleaved by MluI (Supplemental Fig. 3e). Therefore, it is not amplified in the following PCR steps and does not contribute to the results. Nucleosome methylation experiments were performed in a substrate-competitive manner with unmodified and H3K$_C$4me3 or H3K$_C$36me3 nucleosomes mixed in the same reaction. For concentration determination and adjustment, the preparations of the unmodified, H3K$_C$4me3, and H3K$_C$36me3 nucleosomes were analyzed by western blot and GelRed staining for their protein and DNA content (Supplemental Fig. 3c, d). Then, mixtures of unmodified and H3K$_C$4me3 or H3K$_C$36me3 nucleosomes at comparable concentrations were prepared in methylation buffer containing AdoMet, and the methyltransferases were added (Supplemental Figs. 3f, 5b). After the incubation period, the samples were treated with proteinase K to stop the methylation reaction and ensure that the methylated DNA fragments are accessible for the following steps (Supplemental Fig. 3e). Bisulfite conversion was performed followed by a two-step PCR using bisulfite specific primers adding a second barcode in the first step and primers adding indices in the second step. Different libraries were pooled and sequenced by NGS at great depth (Supplemental Table 1). Additionally, methylation reactions with free DNA (without MluI digestion) and bisulfite conversion control reactions treated identically but without enzyme were performed (Supplemental Table 1).

**DNMT3A2 methylates nucleosomal DNA almost exclusively in the linker DNA region.** As described above, the cryo-EM structure of a heterotetrameric DNMT3A/3B3 complex bound to a nucleosome has revealed that one DNMT3B3 subunit binds with two arginine residues to the acidic patch of the nucleosome disc face. Sequence alignments show that the DNMT3B3 residues, which interact with the nucleosome acidic patch, are conserved between DNMT3A and DNMT3B (Fig. 2a), suggesting that a DNMT3A homotetramer could form a similar structure. Therefore, we first used DNMT3A2 and DNMT3AC homotetramers for nucleosome methylation experiments. Unmodified nucleosomes were methylated by DNMT3A2 in several independent reactions using two different preparations of nucleosomes (Fig. 1c and Supplemental Fig. 4a). Strong methylation with methylation levels between 30 and 50% was observed at the four most highly methylated CpG sites in the linker DNA region (position 32–57), whereas little methylation activity was detected at the CpGs within the DNA region that is bound to the nucleosomes (starting at position 65). Strong methylation of site 57 is in agreement with previous data[16]. This characteristic pattern was also observed with H3K$_C$4me3 and H3K$_C$36me3 containing nucleosomes (Figs. 3a, 4b), as well as in methylation reactions with DNMT3AC (Supplemental Fig. 4c). Control reactions with free DNA showed

a more uniform methylation confirming that the nucleosomes remained stable throughout the reaction (Fig. 1c). These distinct methylation patterns of nucleosomal and free DNA were observed in all DNMT3A2 reactions, which were, therefore, averaged and normalized yielding two combined data sets. As expected, control reactions without enzyme consistently revealed low (<0.3%) methylation confirming a high efficiency of the bisulfite treatment (Supplemental Fig. 4b). Deviations from the efficient methylation of linker DNA and protection of the nucleosomal bound CpG sites against methylation were observed at only two sites. The CpG site at position 61 was barely methylated despite being located outside of the 601 binding sequence, which can be explained by steric constraints (see below). In addition, the site at position 189 consistently showed about 15% methylation, although it is located approximately 20 bp inside of the 601 sequence. This finding is in agreement with the known ability of nucleosomes formed on the 601 sequence to transiently unwrap the linker DNA asymmetrically from the right side[41].

**Free and linker DNA methylation follows the flanking sequence preferences of DNMT3A.** Previous work of our group showed that DNMT3A methylates DNA with strong flanking sequence preferences[10,22,42,43]. Comparison of the observed methylation pattern on free DNA with the DNMT3A flanking sequence preferences for the individual CpG sites[10] revealed that relative methylation levels of most CpG sites correspond to the flanking sequence preference (Fig. 1d). Comparison with the profiles of the methylated nucleosomes indicates a very good correlation of the methylation levels of the first 5 CpGs to the flanking sequence preferences suggesting that the main factor determining the methylation activity at these sites in the free linker DNA is the local sequence preference. Within the nucleosomal binding site, no correlation was observed between residual methylation levels and flanking preferences.

**Methylation of nucleosomal DNA by the DNMT3AC/3B3C heterotetramer.** Next, we aimed to replicate our experiments with a similar heterotetrameric enzyme complex as used in the cryo-EM structure. To this end, we employed a double-tag affinity purification strategy (Supplemental Fig. 5a) to generate heterotetrameric complexes containing DNMT3AC and DNMT3B3 C-terminal domain (DNMT3B3C) with the desired stoichiometric composition (Supplemental Fig. 5b). Since the purified protein contained residual amounts of TEV protease, we tested the effect of this enzyme on the recombinant nucleosomes. As shown in Supplemental Fig. 5c, prolonged incubation of nucleosomes with TEV protease did not lead to proteolytic cleavage of the histone proteins, demonstrating the suitability of the purified DNMT3AC/3B3C complex in our standard methylation reaction pipeline. In five independent repeats of the nucleosome methylation with the purified DNMT3AC/3B3C complex, a very similar average methylation pattern was observed as with DNMT3A2 (Fig. 1c, d).

**Methylation levels of CpG sites at positions 57 and 61 are in agreement with the cryo-EM structure.** The two CpG sites at positions 57 and 61 in the linker DNA of our mononucleosome overlap with the corresponding sites in the linker DNA of the cryo-EM structure (Figs. 1a, 2b). Both sites are equally preferred by DNMT3A in free DNA (Fig. 2c). In contrast, the CpG at position 57 shows considerably higher methylation in nucleosomal methylation reactions with DNMT3A2 and the DNMT3AC/3B3C complex compared to free DNA, while methylation of the site at position 61 is almost zero and lower than observed in free

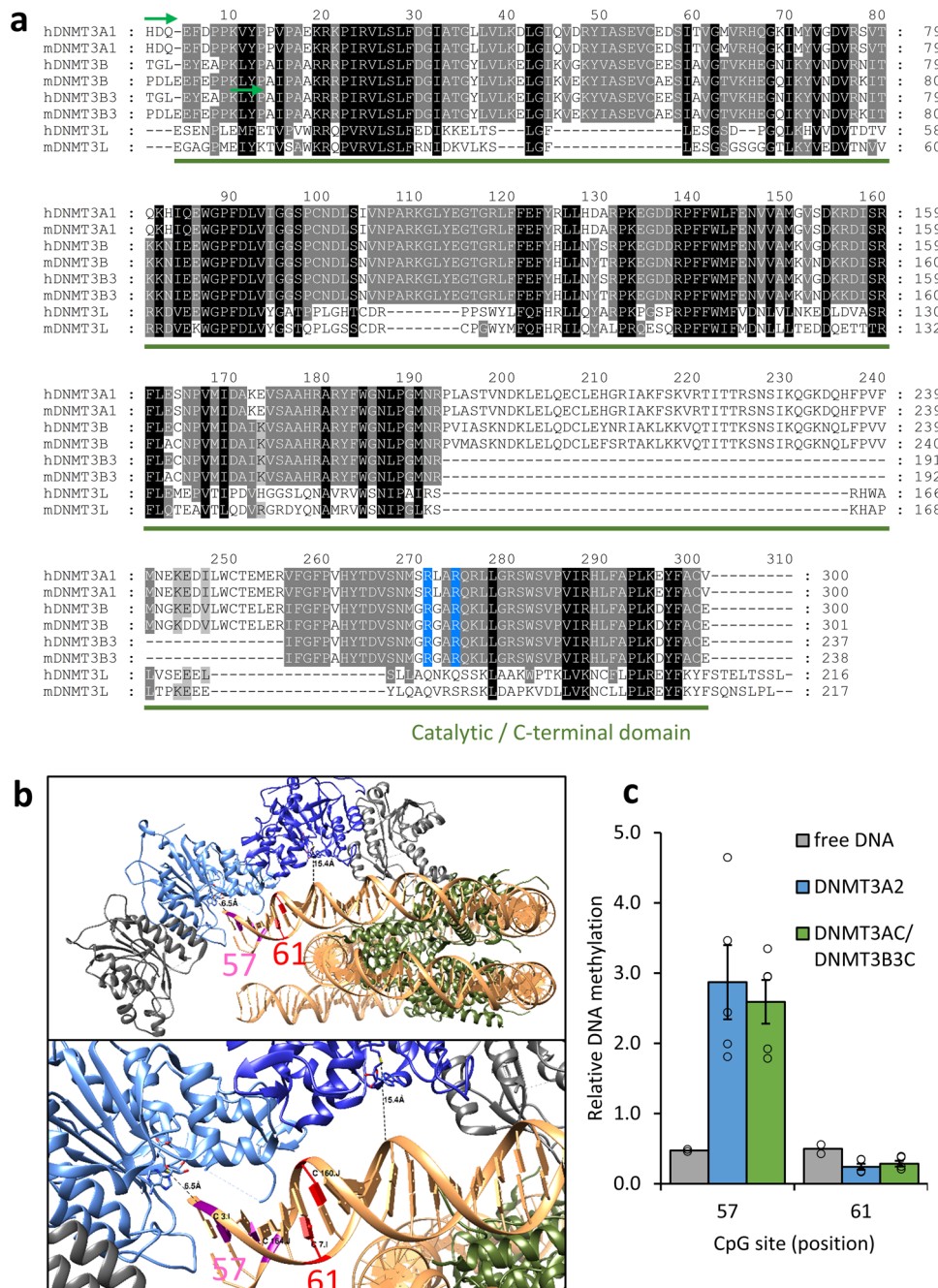

**Fig. 2 Analysis of the linker DNA methylation pattern in the context of structural data. a** Sequence alignment of human and mouse DNMT3A, DNMT3B, DNMT3B3, and DNMT3L showing the C-terminal parts of the enzymes. The first residues resolved in the DNMT3A2/3B3-nucleosome structure are labeled by green arrows. The key DNMT3B3 residues interacting with the nucleosome acidic patch (R740 and R743, highlighted in blue) are conserved between DNMT3A and DNMT3B. The full alignment is shown in Supplemental Fig. 1. **b** Overview of the cryo-EM structure of the DNMT3A/3B3-nucleosome complex (pdb 6PA7). The distal and proximal DNMT3A subunits are colored light and dark blue, the outer DNMT3B3 subunits are colored gray. CpG sites 57 and 61 are marked in purple and red. The lower panel shows an enlarged view focusing on the 10 bp linker DNA. **c** Comparison of the relative methylation levels of CpG sites 57 and 61 in the nucleosome methylation experiments with DNMT3A2 and DNMT3AC/3B3C, as well as in free DNA methylation by DNMT3A2. Error bars represent the SEM if $n \geq 3$.

DNA (Fig. 2c). The increase in methylation activity at site 57 can be explained by the cryo-EM structure of the DNMT3A2/3B3-nucleosome complex, because it approaches the active site of the distal DNMT3A2 subunit by 6.5 Å (SAH-Sulfur to backbone phosphorous atoms distance), which can be easily bridged by base flipping making it a prime target for methylation (Fig. 2b). Conversely, the site at position 61 is located in between the active centers of the proximal and distal DNMT3A2 subunit, but not

close enough to any of them, thereby rendering it inaccessible for catalysis. Hence, the methylation levels of both sites agree very well with intricate details of the cryo-EM structure.

**Trimethylation of H3K4 modulates the methylation pattern of nucleosomes by DNMT3A2.** We next aimed to study the methylation of nucleosomes containing the histone H3K4 and H3K36 lysine trimethylation marks which are known to interact

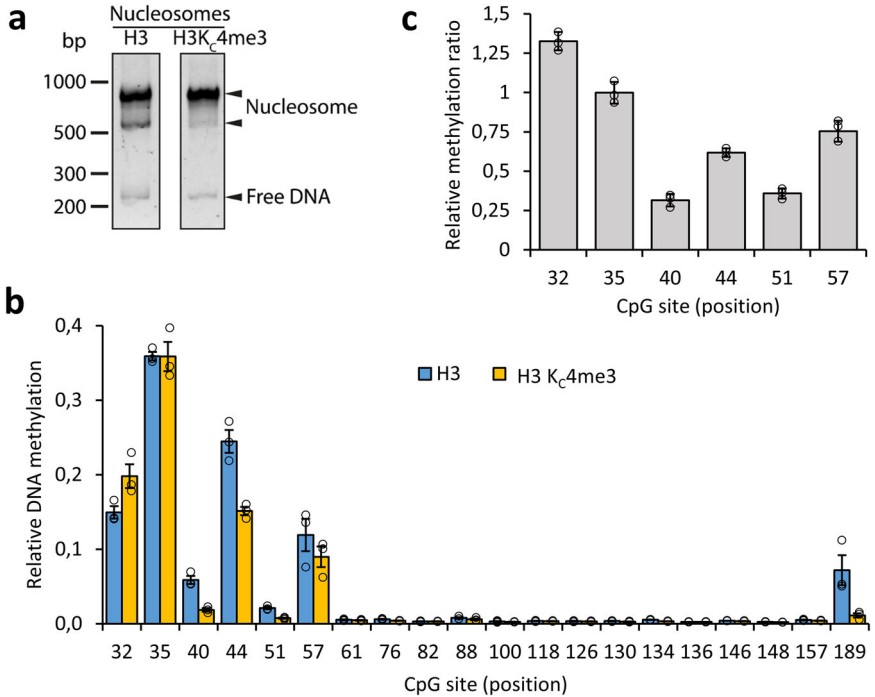

**Fig. 3 Effect of H3K$_C$4me3 on the methylation of nucleosomal DNA by DNMT3A2. a** Gel shift assay of reconstituted unmodified and H3K$_C$4me3 containing nucleosomes. Successful nucleosome reconstitution is confirmed by the strong shifted DNA bands. **b** Relative methylation of each CpG site observed in three independent replicates of the competitive methylation of unmodified and H3K$_C$4me3 nucleosomes. Error bars represent the SEM. **c** Ratio between the competitive methylation levels of H3K$_C$4me3 and unmodified nucleosomes for the CpG sites located in the linker DNA. Note that H3K$_C$4me3 reduced methylation of the 40-57 sites.

with DNMT3A2[4]. The H3 N-terminal tail has been shown to bind to the ADD domain of DNMT3A2 and thereby allosterically stimulate its catalytic activity, but this binding is inhibited by di- and trimethylation of K4[16,32,35]. To study this effect in a nucleosomal context and on a quantitative basis, we generated nucleosomes containing H3K$_C$4me3 (Fig. 3a), and performed methylation experiments with them. To allow for reliable quantification of the methylation levels of different nucleosomes, methylation reactions were performed using mixtures of H3K$_C$4me3 containing and unmodified nucleosome substrates in competition. The resulting methylation patterns showed overall similarity between the modified and unmodified nucleosomes (Fig. 3b), because methylation largely occurred in the linker DNA region. However, the CpGs close to the nucleosome core (sites 40, 44, 51, and 57) showed a reduction in methylation rates of 25–70% (Fig. 3c) on nucleosomes containing H3K$_C$4me3 when compared to unmodified nucleosomes. The most distal CpG sites (32 and 35), however, were unaffected or even showed slightly increased methylation.

**DNMT3A2 methylates nucleosomal DNA with a preference for H3K$_C$36me3.** H3-tails di- or trimethylated at K36 are known to bind to DNMT3A2 at the PWWP domain and stimulate its activity[23,24,28]. Therefore, methylation reactions were conducted with nucleosomes containing the trimethyllysine analog at position K36 (Fig. 4a) in competition with unmodified nucleosomes. The overall methylation pattern of H3K$_C$36me3 nucleosomes was identical to unmodified nucleosomes with a strong preference for the first six and the last CpG sites (Fig. 4b). However, in five independent reactions, a 1.5-fold higher methylation of the CpG sites in the linker DNA regions was observed for nucleosomes containing H3K$_C$36me3 (Fig. 4c). Methylation reactions of unmodified and H3K$_C$36me3 containing nucleosomes with the

catalytic domain of DNMT3A revealed similar methylation pattern as the reactions with DNMT3A2 (Fig. 4d). However, unexpectedly, competitive nucleosome methylation using DNMT3AC showed an even stronger 3.6-fold stimulatory effect of H3K$_C$36me3 (Fig. 4c, d), although the PWWP domain is missing in this context. These data indicate that H3KC36me3 stimulates DNMT3A in a previously undescribed mechanism independent of its interaction with the PWWP domain.

**Binding dynamics of the H3-tail influenced by H3K36 trimethylation.** The PWWP domain independent stimulation of DNA methylation by H3K$_C$36me3 could be explained by weakening of the H3-tail binding to the linker DNA, which would increase the accessibility of the linker DNA for methylation. Quenching of tryptophan fluorescence upon DNA binding is a well-established method to study protein–DNA interactions[44,45]. We, therefore, aimed to investigate the influence of H3K$_C$36me3 on the dynamic interaction of the H3-tail with the linker DNA by fluorescence spectroscopy. To this end, we introduced an A15W mutation into the H3-tail as a fluorophore (nucleosomes do not contain additional tryptophan residues) and generated recombinant nucleosomes containing this mutation in combination with or without the H3K$_C$36me3-analog. Successful nucleosome reconstitution was documented by EMSA (Supplemental Fig. 6a, b) and equal protein amounts confirmed by western blot analysis (Supplemental Fig. 6c). Then, we measured the tryptophan fluorescence emission spectra of the different nucleosomes. In three independent experiments, the fluorescence of the nucleosomes containing the H3K$_C$36me3-analog was on average 1.5 times higher than the fluorescence of the unmodified nucleosomes (Supplemental Fig. 6d) suggesting that the presence of H3K$_C$36me3 reduces the binding of the H3-tail to the linker DNA (Supplemental Fig. 6e).

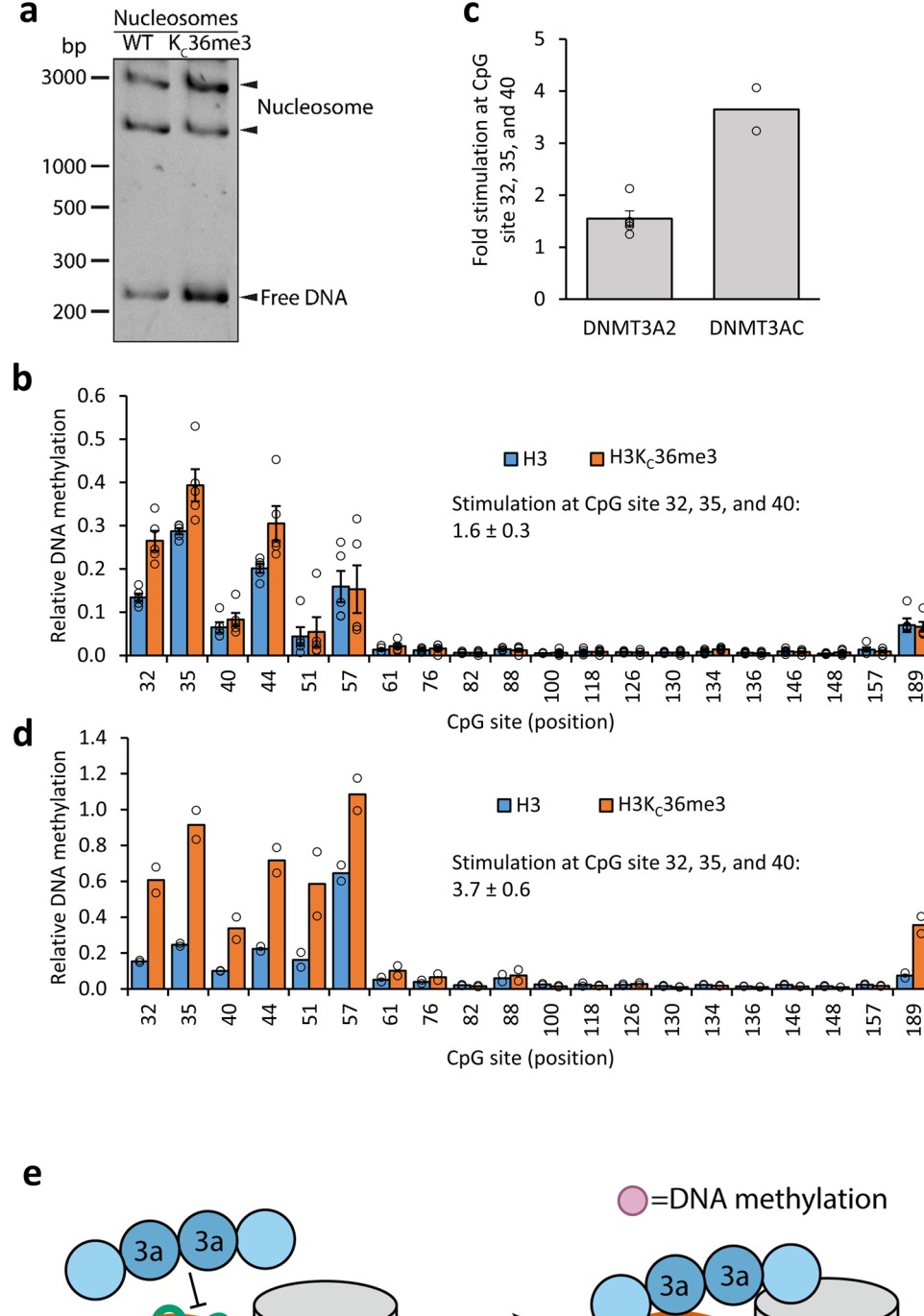

**Fig. 4 H3K$_C$36me3 recognition and details of the methylation pattern on the linker DNA. a** Gel shift assay of unmodified and H3K$_C$36me3 containing nucleosomes. Successful nucleosome reconstitution is confirmed by the prominent shifted DNA bands. **b** Relative CpG site methylation levels observed in five independent competitive methylation experiments of unmodified and H3K$_C$36me3 nucleosomal substrates with DNMT3A2. For comparison, the activities in the independent reactions were normalized to the average activity on unmodified nucleosomes. Error bars represent the SEM. **c** Ratio of the methylation levels of unmodified and H3K$_C$36me3-containing nucleosomes for the three most strongly methylated CpGs with DNMT3A2 and DNMT3AC. H3K$_C$36me3 nucleosome methylation is 1.5-fold increased in the case of DNMT3A2 ($p$-value: $1.15 \times 10^{-4}$, based on a two-sided T-Test assuming equal variance). For the DNMT3AC, the ratio is more than 3.6-fold increased ($p$-value: $2.32 \times 10^{-3}$, based on a two-sided T-Test assuming equal variance). Error bars represent the SEM if $n \geq 3$. **d** Relative CpG site methylation levels observed in two independent replicates of the competitive methylation experiment of unmodified and H3K$_C$36me3 modified nucleosomes with DNMT3AC. **e** Schematic model of the inhibition of linker DNA methylation by H3-tail binding and the reduction of the H3-tail interaction with linker DNA by H3K36me3.

## Discussion

Our data reveal that linker DNA is readily methylated by DNMT3A2, DNMT3AC/3B3C, and DNMT3AC, but nucleosomal DNA is highly protected against methylation, which is in general agreement with previous in vitro methylation data[16,17,46], with the low activity of DNMT3A observed on core nucleosomes lacking linker DNA[47], and inhibition of DNMT1 on nucleosomal DNA[48]. Reduced methylation of nucleosomal DNA has been observed under specific experimental settings focusing on DNMT3A activity in vivo[27], but in general cellular DNA is equally or even higher methylated at nucleosome binding sites when compared with linker DNA[18,49]. This finding can be explained by the activity of remodeling factors that have been shown to be essential for DNA methylation like Ddm1[50–52], HELLS[53,54], or SWI/SNF[55].

For the comparison of the DNA methylation data with the cryo-EM DNMT3A2/3B3-mononucleosome complex structure[36], the two CpG sites at positions 57 and 61 on linker regions of the nucleosomal substrates are particularly interesting, as they are present in the structure as well. The acidic patch interaction of the DNMT3B3 subunit anchors the central catalytic DNMT3A2 domains in a way that site 57 is ideally positioned to interact with the active site of one DNMT3A2 subunit, while site 61 is placed in between the active centers of both DNMT3A2 subunits and cannot interact with any of them. Our kinetic analyses with DNMT3A2 and DNMT3AC/3B3C revealed high methylation of site 57, but low methylation of site 61. This finding is perfectly matching with the DNMT3A2/3B3-nucleosome structure, and it suggests that the DNMT3A2 tetramer contacts the acidic patch in a similar way as the DNMT3A2/3B3 complex in agreement with the strong conservation of the residues that are involved in the binding. Moreover, our data strongly suggest that the cryo-EM DNMT3A2/3B3-nucleosome structure captures the complex in a catalytically competent conformation.

However, it is interesting to note that the strongest methylation in our experiment was observed in the first CpG sites of our construct located up to 33 bp away from the nucleosome entry/exit site, which was not included in the cryo-EM structure. Different scenarios for the strong methylation of these sites are possible: Binding of the DNMT3A2 tetramer to these loci could occur without an interaction with the nucleosome core, in a manner described earlier for the interaction with free DNA[8,9] or by binding of a second tetramer side-by-side with the anchored complex as described recently[56]. On the other hand, it also appears possible for the complex to remain in the nucleosome-bound state and exploit the flexibility of the linker DNA and some hinge movement at the nucleosome binding site and perhaps within the DNMT3A2 tetramer to access the more remote sites. Future studies will be needed to resolve the processes leading to the highly efficient methylation of CpG sites away from the nucleosome core.

DNMT3A exists in an autoinhibitory conformation, which is relieved upon binding of the H3-tail peptide to its ADD domain. Trimethylation of the H3K4 residue suppresses this binding, leading to impaired catalytic activity[35]. We investigated the effect of H3K$_C$4me3 on nucleosome methylation by DNMT3A2 and showed reduced methylation at CpG sites close to the nucleosome core but equal or even slightly elevated activity at sites further away. This result suggests that H3-tail binding to DNMT3A2 specifically stimulates the methylation of linker DNA sites, which are close enough to the nucleosome core to allow simultaneous interaction of DNMT3A2 with the CpG site and the H3-tail. Trimethylation of H3K4 hinders this activation process resulting in a lower methylation of the corresponding CpG sites in nucleosomes containing H3K$_C$4me3.

In addition, our data revealed an increase in methylation of H3K$_C$36me3 containing nucleosomes by DNMT3A2 but unexpectedly also by DNMT3AC. This effect was observed in a competitive methylation experiment of unmodified and H3K$_C$36me3 modified nucleosomes, ruling out any effects unrelated to the H3K$_C$36me3 modification. The trimethyllysine-analog is chemically identical to the trimethyllysine side-chain, except that it contains a sulfur atom instead of a methylene group at the γ-position. While it is known that it does not fully mimic trimethyllysine under all conditions, binding of H3K$_C$36me3 containing peptides had been shown previously for different PWWP domains, e.g., in hMSH6[57], DNMT3A[58], and LEDGF[59] suggesting that it is suitable for our study. As DNMT3AC does not carry the ADD and PWWP domains, this observation indicates that both of these domains are not involved in the stimulation by H3K$_C$36me3, in particular, that the effect is independent of H3K36me2/3 binding to the PWWP domain.

To explain this observation, it must be considered that several studies using NMR, chemical reactivity, molecular dynamics, and fluorescence analyses showed that the H3-tail dynamically interacts with the linker DNA ultimately even leading to a transient wrapping of the H3 around the DNA[60–66]. Certainly, binding of the polycationic H3-tail to the linker DNA would inhibit the linker DNA interaction with DNMT3A and thereby reduce DNA methylation (Fig. 4e). Lysine 36 is positioned at the basis of the H3-tail next to the linker DNA entry/exit site from the nucleosome and the positive charge of its side chain leads to a strong interaction with the DNA, which can anchor the entire H3-tail on the linker DNA and stabilize the DNA-bound conformation of the H3-tail. The K36-DNA interaction is expected to be reduced by trimethylation of K36, because all H-bond options of the amino group are blocked by methylation. Moreover, the fit of the H3-tail into the minor groove of the DNA is reduced after lysine trimethylation. This weakens the H3-tail-linker DNA interaction, which could explain the direct stimulation of DNA methylation by H3K36me3 (Fig. 4e). In agreement with this model, we observed a reduction of the H3-tail interaction with linker DNA by H3K$_C$36me3 in fluorescence spectroscopy experiments. The stronger H3K$_C$36me3 stimulation of DNMT3AC, when compared with DNMT3A2, is in agreement with this model, because in DNMT3A2 the PWWP and ADD domains can interact with the H3-tail and suggesting that this may help to keep it away from the linker DNA. These options are not available in DNMT3AC, explaining why this enzyme is more dependent on the presence of H3K36me3 to get access to the linker DNA. In summary, our data demonstrate that the H3-tail interaction with the linker DNA reduces DNMT3A activity. This interaction can be weakened by trimethylation of H3K36 leading to an increase in methylation activity of DNMT3A.

H3K36 methylation in the bodies of active genes is among the evolutionary most ancient histone methylation and widely found in many species[67]. It recruits different silencing factors including HDACs and DNMTs and prevents inappropriate transcriptional initiation in gene bodies[67,68]. Similarly, CpG methylation on gene bodies of expressed genes is a conserved feature of eukaryotic genomes[69–71], and the targeting role of H3K36 methylation for DNA methylation in gene bodies[27], but also in intergenic regions[28], is deeply rooted in the eukaryotic phylogeny[71]. However, the reasons for this tight and conserved connection of these particular modifications are unknown. Our data suggest an evolutionary model, in which the direct functional effects of H3K36 methylation may have preceded its signaling role. Initially, H3K36 methylation might have been deposited in gene bodies of active genes, because of its ability to release H3-tails from the linker DNA and thereby support transcription. Later, the H3K36 methylation mark was connected with silencing effects to prevent

inappropriate transcriptional initiation in gene bodies. The intrinsic capacity of H3K36 methylation to stimulate DNA methylation could have triggered the deposition of DNA methylation leading to the appearance of gene body methylation. Later, this effect was supported by a signaling role of H3K36 methylation in the recruitment of DNMTs and other silencing factors mediated by specific reading domains, like PWWP domains. Hence, this model proposes that the direct stimulatory effects of H3K36 methylation on DNA methylation preceded its signaling function in the recruitment of DNMTs, which immediately explains why both modifications became functionally connected. In agreement with this scenario, sequence alignments revealed that plant DNMT3 related enzymes (the DRM family of DNA methyltransferases[72]) do not contain a PWWP domain, while PWWP domains are found in vertebrate and invertebrate DNMT3 enzymes[73], suggesting that PWWP domains were added to the *DNMT3* gene family after the separation of the plant and animal lines.

In conclusion, we observed protection of the nucleosomal DNA against methylation by DNMT3A, but efficient methylation of nucleosomal linker DNA next to the nucleosome core, but also away from it. Methylation of linker regions and free DNA followed known sequence preferences of DNMT3A. Methylation levels of CpG sites next to the nucleosome are in agreement with the DNMT3A-mononucleosome structure supporting the notion that the structure captured a catalytically active conformation. H3K4 methylation reduces DNA methylation of CpG sites close to the nucleosome core. Strikingly, our data demonstrate a direct stimulatory effect of H3K36me3 on linker DNA methylation which is independent of the DNMT3A-PWWP domain and can be explained by H3K36me3 reducing the interaction of H3-tails with linker DNA. Based on this, we propose an evolutionary model in which the direct stimulatory effects of H3K36me3 on DNA methylation preceded its signaling function explaining the wide distribution of the "active gene body-H3K36me3-DNA methylation" connection.

## Materials and methods

**DNMT3A2 and DNMT3AC overexpression and purification**. The murine full-length DNMT3A2 construct was transformed into BL21-CodonPlus cells using the heat shock method and the cells were transferred to selective agar plates. Single colonies were used to inoculate a preculture, which was grown for 7 h at 37 °C. The preculture was then transferred to a 1 L main expression culture, which was grown to an $OD_{600}$ of 0.6–0.8 at 37 °C with shaking. At this point, protein expression was induced by the addition of IPTG to a final concentration of 500 µM, and the cells were cultivated with shaking at 20 °C for 12 h. The cells were harvested by centrifugation at 5000 rcf for 15 min. The cell pellet was resuspended in sonication buffer (30 mM $KH_2PO_4/K_2HPO_4$ pH 7.0, 500 mM KCl, 1 mM EDTA, 0.2 mM DTT, 10% glycerol, 20 mM imidazole) and the cells were disrupted by sonication (20 cycles with 15 s active sonication and 30 s pause, 50% amplitude, Epishear Active Motif). The cell lysate was centrifuged at 40,000 rcf for 90 min and the supernatant was loaded onto a Ni-NTA column using the NGC FPLC system (Bio-Rad) which was equilibrated with sonication buffer. After washing with 50 mL of sonication buffer, the protein was eluted with sonication buffer containing 220 mM imidazole. Fractions were collected, pooled according to yield and purity, and load on a Superdex 200 16/60 PG gel filtration column equilibrated in dialysis buffer (20 mM HEPES pH 7.5, 200 mM KCl, 0.2 mM DTT, 1 mM EDTA, 10% glycerol). The sample was eluted isocratically, fractions were collected, pooled according to purity, and concentrated tenfold using Amicon Ultra-4 centrifuge filters (30 kDa cutoff, Merck Millipore). Aliquots were flash-frozen in liquid $N_2$ and stored at −80 °C. Expression and purification of DNMT3AC were performed as described[10,43].

**Purification of DNMT3AC/3B3C heterotetramers**. The bacterial expression constructs encoding for the MBP-TEV-DNMT3AC, as well as the His-tagged murine DNMT3B3C were used for overexpression of the proteins in *E.coli* following the same procedure as described earlier[42]. In order to generate heterotetramers consisting of DNMT3AC and DNMT3B3C, a two-step purification method was applied similarly as described previously[42], but including the proteolytic removal of the MBP-tag. In brief, after cell lysis, the solutions containing the two different proteins were mixed, incubated for 30 min on ice, and batch-incubated with amylose-linked agarose beads for 1 h at 8 °C. The beads were washed two times with sonication buffer and the MBP-tagged protein was cleaved by treatment with TEV protease for 30 min at 20 °C. The resulting eluate was passed over Ni-NTA-agarose beads, washed two times with sonication buffer, and eluted with sonication buffer containing 220 mM imidazole. Aliquots were flash-frozen in liquid $N_2$ and stored at −80 °C.

**Histone overexpression and purification**. Purification of individual histone proteins was performed essentially as described with some adaptations[74]. Briefly, pET21a expression constructs containing the corresponding histone genes (H3.1, H4, H2A, and H2B) were transformed into BL21-Codon Plus cells using the heat shock method and the cells were grown on agar plates overnight. Single colonies were used to inoculate an overnight preculture, which was grown at 37 °C. The preculture was subsequently used to inoculate the main culture with a volume of 500 mL, which was grown to an $OD_{600}$ of 0.6–0.8 at 37 °C. Protein expression was induced by the addition of IPTG to a final concentration of 1 mM followed by incubation of the cell culture with shaking for 3 h at 20 °C. Cells were harvested by centrifugation at 5000 rcf for 15 min, washed using STE buffer (10 mM Tris/HCl pH 8, 100 mM NaCl, and 1 mM EDTA), pelleted again by centrifugation at 5000 rcf for 15 min and stored at −20 °C.

The bacterial pellets were thawed and resuspended in SAU buffer (10 mM Sodium acetate pH 7.5, 1 mM EDTA, 10 mM Lysine, 5 mM β-mercapthoethanol, 6 M Urea, and 200 mM NaCl). Cells were disrupted by sonication (20 cycles with 15 s active sonication and 30 s pause, 50% amplitude, Epishear Active Motif). The lysate was cleared by centrifugation at 40,000 rcf for 1 h and passed through a 0.45 µM syringe filter (Chromafil GF/PET 45, Macherey-Nagel). The strong cation exchange column HiTrap SP HP (5 mL, GE Healthcare) was connected to an NGC FPLC system (Bio-Rad) and equilibrated with 5 column volumes (CV) of SAU buffer. The lysate was passed through the column and the column was washed with 8 CV of SAU buffer. Proteins were eluted using a salt gradient from 200 mM to 800 mM NaCl. Fractions were collected, pooled according to purity and yield, dialyzed against pure water with two changes overnight, and dried in a vacuum centrifuge for storage at 4 °C.

**Installation of trimethyllysine analogs**. For the generation of a trimethyllysine-analog, the previously described nucleophilic exchange via a cysteine residue was applied[39]. In order to specifically generate the trimethyllysine analog at position 4 or 36 of H3, the lysine to cysteine mutation was introduced by PCR-based site directed mutagenesis[75]. Using the same approach, the two other cysteine residues in the H3 protein (C96 and C110) were mutated to serine to prevent the unwanted conversion of these residues. The dried H3K4C or H3K36C proteins were dissolved in alkylation buffer (1 M HEPES pH 7.8, 4 M guanidinium chloride, 10 mM methionine). DTT was added to a final concentration of 20 mM and the protein was reduced for 1 h at 37 °C. Then, 100 mg/mL 2-bromoethyltrimethylammoniumbromide were added to the solution and incubated for 2 h at 37 °C. The reaction was quenched by the addition of 50 µL pure β-mercapthoethanol per mL of solution and incubation at room temperature for 30 min. The sample was dialyzed against pure water (with two changes) and dried in a vacuum centrifuge for storage at 4 °C. Successful conversion of the cysteine into the trimethyllysine-analog was confirmed by MALDI mass spectrometry.

**Refolding of histone octamers**. For the reconstitution of histone octamers, a previously described protocol was used[40]. The individual histone proteins were dissolved in unfolding buffer (20 mM Tris/HCl pH 7.5, 7 M guanidinium chloride, 5 mM DTT) and their concentrations determined spectrophotometrically. The proteins were combined in a ratio of 1 (H3, H4) to 1.2 (H2A, H2B) and dialyzed against two charges of refolding buffer (10 mM Tris/HCl pH 7.5, 1 mM EDTA, 2 M NaCl, 5 mM β-mercapthoethanol) overnight. To isolate the octamers from the refolding mixture, the sample was separated by size exclusion chromatography using a Superdex 200 16/60 PG column equilibrated in refolding buffer. Fractions were collected, pooled according to purity and concentrated tenfold using Amicon Ultra-4 centrifuge filters (30 kDa cutoff, Merck Millipore). Samples were flash frozen in liquid $N_2$ and stored at −80 °C.

**Nucleosome reconstitution**. To generate a DNA binding fragment for nucleosome reconstitution with flexible barcodes, the DNA sequence variant of the Widom 601 sequence[38] that was used in the cryo-EM structure of DNMT3A2 bound to a mononucleosome[36] together with a CpG-rich linker sequence was cloned into a TOPO-TA vector. The MluI restriction site was introduced into the 601 sequence by site-directed mutagenesis[75]. Large quantities of nucleosome binding DNA were amplified by PCR using barcoded primers specific for each nucleosome variant. DNA and histone octamers were combined in different ratios ranging from equimolar to twofold octamer excess. These samples were transferred to Slide-A-Lyzer microdialysis devices (ThermoFisher) and dialyzed against high salt buffer (10 mM Tris/HCl pH 7.5, 2 M NaCl, 1 mM EDTA, 1 mM DTT), which was continuously replaced by two liters of low salt buffer (same composition as high salt but with 250 mM NaCl) over the course of 24 h. Afterwards, the samples were further dialyzed against storage buffer (10 mM Tris/HCl pH 7.5, 1 mM EDTA, 1 mM DTT, and 20% glycerol) overnight, flash-frozen in liquid $N_2$, and stored at −80 °C.

**Methylation experiments**. For the competitive nucleosome methylation experiments, 0.6 pmol of each nucleosome variant were digested with MluI (NEB) for 60 min at 37 °C in 10 µL NEB Cutsmart buffer (50 mM KOAc/20 mM Tris-acetate pH 7.9, 10 mM Magnesium Acetate, 100 µg/mL BSA) to remove residual unbound DNA. Afterwards, DNMT3A2 or DNMT3AC was added to the mixture to a final concentration ranging from 0.5 to 3 µM and in 80 µL NEB Cutsmart buffer supplemented with 10 mM EDTA and 25 µM AdoMet (Sigma). The methylation reaction was allowed to proceed for 2 h at 37 °C. To stop the reaction and remove all nucleosome-bound proteins, proteinase K was added to the reaction and the sample was incubated for further 60 min at 37 °C. The resulting unbound DNA was purified from the reaction mixture using the Nucleospin Gel and PCR cleanup kit (Macherey-Nagel). Bisulfite conversion of the methylated DNA was performed using the EZ DNA Methylation-Lightning kit (Zymo Research). Methylation of free DNA was conducted the same way using 15 µM DNA.

**Library preparation and sequencing analysis**. Sample-specific barcodes and indices were added to the DNA by PCR amplification in a two-step PCR process. Briefly, in the first PCR, barcoded primers were used to amplify the bisulfite converted nucleosome DNA using the HotStartTaq Polymerase (Qiagen) and the resulting 321 bp fragment was purified using the Nucleospin Gel and PCR cleanup kit (Macherey-Nagel). In the second PCR step, adaptors and indices required for sequencing were added by amplification with the respective primers and the Phusion polymerase (ThermoFisher). The final 390 bp product was purified and used for Illumina paired end 2×250 bp sequencing. Datasets were analyzed using a local instance of the Galaxy bioinformatics server[76]. Sequence reads were trimmed with the Trim Galore! Tool (developed by Felix Krueger at the Babraham Institute) and subsequently paired using PEAR[77]. The reads were filtered according to the expected DNA length using the Filter FASTQ tool and mapped to the corresponding reference sequence using bwameth to determine the percentage of methylated CpGs[78,79]. All statistical analyses were done in Microsoft Excel 2016.

**Nucleosome tryptophan fluorescence emission experiments**. Recombinant nucleosomes containing the H3A15W mutation with or without H3K36$_C$me3 were prepared by introduction of the A15W mutation into the expression constructs of histone H3.1 and histone H3.1 K36C, followed by protein purification and subsequent conversion of K36C into the trimethyllysine analog. The nucleosomes with and without H3K36$_C$me3 were analyzed by Western Blot against Histone H3 (ab-1791, abcam) to ensure that equal concentrations were used. For the fluorescence spectroscopy experiments, the nucleosomes were diluted in storage buffer to a final concentration of 100 nM. Fluorescence measurements were conducted in a Jasco FP-8300 spectrofluorometer using $\lambda_{Ex} = 280$ nm, $\lambda_{Em} = 300$–450 nm, excitation bandwidth = 5 nm, emission bandwidth = 10 nm, scan speed = 200 nm/min, sensitivity = medium, response = 0.5 s, 5 accumulations at 20 °C.

**Statistics and reproducibility**. The number of independent experimental repeats is indicated for each experiment. *P*-values were determined using two-sided T-Test assuming equal variance.

**Reporting summary**. Further information on research design is available in the Nature Research Reporting Summary linked to this article.

## Data availability

The processed sequencing data have been deposited in the data repository of the University of Stuttgart DARUS (https://darus.uni-stuttgart.de/) under https://doi.org/10.18419/darus-1252. Source data and uncropped images are provided in Supplemental Data 1 and Supplemental Fig. 7. All other data are available from the corresponding author upon reasonable request.

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

## Acknowledgements

We acknowledge support by the Max Planck-Genome-Center Cologne in NGS sequencing. This work was supported by the Deutsche Forschungsgemeinschaft DFG (grant JE 252/10 to A.J.).

## Author contributions

A.B. and M.D. conducted all experiments with support from T.S., N.G., F.D., E.M., and S.A. P.B. supervised the sequencing data analysis. A.B. and A.J. prepared the manuscript draft and figures. A.J. devised and supervised the work. All authors contributed to data interpretation and editing of the manuscript. The final manuscript was approved by all authors.

## Funding

## Competing interests

The authors declare no competing interests.
