## [Peer Review File · Communications Biology]

Reviewers' comments:

Reviewer #1 (Remarks to the Author):

This paper shows that strong linker-DNA methylation was observed next to the nucleosome core, and the stimulatory effect of H3K36me3 on linker DNA methylation is independent of the DNMT3A-PWWP, where H3K36me3 possibly hindered the H3 tail-linker DNA interaction.

Overall, the data in this paper is well represented and is in good agreement with the previous publications and structures. There are two points that need to be specified or clarified:

1. In Supplemental 2A and 2B, the DNA runs at ~200 bp and nucleosome runs between 500-1000 bp. In Supplemental 2C and 2E, the DNA still runs at ~200 bp, while the nucleosome runs between 2000-3000 bp in the native PAA gel. Is it caused by different gel types, or does the labeling needs to be clarified?
2. Since this paper demonstrate a direct stimulatory effect of H3K36me3 on linker DNA methylation independent of the DNMT3A-PWWP domain, and proposed a model of "active gene body-H3K36me3-DNA methylation", there should be a Figure 3 to represent/explain this mechanism/model. Otherwise, it is too abstract for readers to understand the key points.

Reviewer #2 (Remarks to the Author):

In this manuscript, Bröhm and colleagues assessed the methyltransferase activity of either full-length or the catalytic domain of Dnmt3a2 against either free or nucleosome-bound DNA. Two observations were made: (i) Nucleosome-bound DNA is protected from methylation by Dnmt3a2 in a manner consistent with the recently published cryo-EM structure; (ii) H3K36 methylation stimulates the Dnmt3a2 activity independent of its PWWP domain. While these findings (especially the H3K36 methylation part) are potentially interesting, the major concern is the physiological relevance of their in vitro assays.

1. It is well-known that adaptor protein either Dnmt3l or Dnmt3b3 is required for Dnmt3a's optimal activity. Indeed, recent paper (PMID: 33004415) shows that re-expression of Dnmt3a2 in the absence of adaptor proteins has virtually no effect on de novo DNA methylation. The cryo-EM structure that was heavily referred to in the study also involved Dnmt3a2-Dnmt3b3 heterotetramer, not Dnmt3a2 alone. Does Dnmt3a2 form homotetramer? It is highly recommended that the authors repeat the assays with either Dnmt3l or Dnmt3b3 added to make their findings physiologically relevant.
2. The claim that H3K36me3 stimulates Dnmt3a2 activity independent of PWWP domain is interesting. However, this part needs to be strengthened. The conclusion was made based on comparing full-length vs. the catalytic domain of Dnmt3a2. Dnmt3a2 has other regulatory domains (eg. ADD) in addition to PWWP domain. So the better comparison here should be PWWP-deleted or mutated Dnmt3a2 (in complex with Dnmt3l or Dnmt3b3 as discussed above). Also trimethyllysine analog was used to mimic H3K36me3, the study needs to be verified with true H3K36me3-modified nucleosomes.
3. Related to the above point, authors suggest that H3K36me3 may affect DNA-histone interaction, thus allowing better engagement of Dnmt3a2. This suggests that H3K36me3 could act in cis which is a quite provocative idea. Therefore this conclusion needs to be supported by further evidence and structural or biophysical analysis of the H3K36me3 nucleosome.

Reviewer #3 (Remarks to the Author):

The manuscript followed recent cryo-EM structure of Dnmt3a2-3b3 and performed DNA methylation assays of Dnmt3a2 (full length protein and the catalytic domain) in a nucleosomal

context, with and without H3K36 methylation. The authors concluded that (1) the methylation occurs mostly on the linker DNA (where the DNA fragment was not protected by histones), (2) the difference of the DNA methylation level at a given CpG site is largely influenced by neighboring sequence, and (3) H3K36 methylation stimulates DNA methylation by both the full-length enzyme and the catalytic domain, regardless of whether the histone binding domain (PWWP) existed. Given that the preferential methylation of linker DNA has been abundantly documented, the finding that the H3K36me3 stimulates the methylation of linker DNA is of interest but that essentially is the only new finding presented. There are no new mechanistic or unexpected insights and the work is limited by the failure to include an accessory protein in the assays given what is known about the essential function of these proteins.

Below are the detailed concerns with this manuscript: (1) the in vitro experimental setting is different from that of the structural data that the authors used to justify their results, and (2) the conclusions are contradictory to what was previously published from the same laboratory.

In cells, Dnmt3a is regulated by Dnmt3L in ES cells or Dnmt3b3 (or other Dnmt3b inactive isoforms) in somatic cells. In the latest cryo-EM structure, Peter Jones and his colleagues used Dnmt3a2-Dnmt3b3 in complex with nucleosome, where Dnmt3b3 is in contact with nucleosome disk surface. However, in the current study, the authors used Dnmt3a2 alone. There is no biophysical data to support Dnmt3a2 alone forms "tetramer". As a matter of fact, a paper from the same lab claimed Dnmt3a forms oligomer (reference 13). Thus, the experiment should be performed using the same configuration of Dnmt3a2-Dnmt3b3.

The main question the authors addressed in the manuscript has been addressed by the same laboratory previously (reference 15). In 2005, they used Dnmt3a full length and the catalytic domain, same as used in the current study, yet the opposite conclusions were made (see below). Although the technique used to detect methylation is different, the principle is the same (bisulfite sequencing), and the same enzymes were used. The differences should be fully addressed. A solid conclusion should not change as a function of time.

Examples of different conclusions made in 2005 (reference 15) and now (current manuscript):

(2005) The methylation sites are not clustered towards the ends of the DNA, which are typically more accessible than the core region of the nucleosome.

(Current) Strong methylation was observed in the linker DNA region, whereas little methylation activity could be observed for the CpGs within the DNA region that is bound to the nucleosomes.

(2005) No significant difference in binding and methylation between all four tail-less histone variants. No difference in the activity of Dnmt3a on H3K9methylated nucleosome.

(Current) Nucleosome containing H3K36methylation analogs stimulates Dnmt3a activity, even under the context of without histone tail binding domain (i.e., Dnmt3a catalytic domain alone).

Is the stimulation effect specific for H3K36 methylation? How about H3K4me3 and H3K9me3 in the absence of histone binding?

(2005) CGG is the preferred site of methylation.

(Current) Figure 1D, please show the sequence information.

Reference 10 showed Dnmt3a has a different preference (TnCG-c/t-c/t).

(2005) Methylation events at non-CpG sites.

(Current) no mention of non-CpG methylation

(2005) methylation activity of Dnmt3 catalytic domain was strongly inhibited by nucleosomal DNA. Binding of Dnmt3a catalytic domain to nucleosome was very weak.

(Current) Do these observations still hold?

Reviewer #4 (Remarks to the Author):

This study investigates the impact of trimethylation of histone H3 at lysine 36 (H3K36me3) on DNA methylation carried out by the methyltransferase DNMT3A2. Using an in vitro methylation assay on reconstituted nucleosomes containing DNA based on the Widom-601 sequence with an extended 5' linker, the authors demonstrate that DNMT3A2 shows a preference for certain CpG sites in a nucleosomal context vs free DNA, and the presence of a histone octamer protects against methylation of octamer-bound DNA. Furthermore, H3K36me3-containing nucleosomes correlate with higher levels of methylation when compared to those containing unmodified H3, and this effect is shown to be more dramatic when the assay is performed using the DNMT3A catalytic domain (DNMT3AC). Finally, using cryo-EM data it is demonstrated that the main site preferentially methylated by DNMT3A2 in the presence of a nucleosome (site 57) is situated close to the catalytic core of one of the DNMT3A2 subunits in the heterotetramer formed by DNMT3A/DNMT3B3 when bound to the nucleosome.

This manuscript is well-written, and the methods used are nicely described and easy to follow. The claims made are for the most part well-supported by the data, and a compelling model is presented for the role and evolution of the relationship between DNA methylation and H3K36me3. There are some (mostly minor) comments/questions outlined below. I would consider points #4 & #6 most important, as they potentially impact the model presented in this article.

Comments/questions:

1. Fig. S1A: perhaps label black line as "H3Kc36me3" instead of "H3K36me3" to make it obvious that this is the converted sample rather than some other H3K36me3 peptide
2. Table S1: DNMT3A2 WT H3Kc36me3 repeats #1 & 2 have far fewer reads than the H3 samples. Do you know why this might be the case?
3. Fig. 1C. What order were the 2 free DNA replicates done relative to the nucleosomal replicates? Two of the nucleosomal replicates at site 57 are quite close to the free DNA replicates. Were these done together and the subsequent (higher) nucleosomal replicates done before/after?
4. The cryo-EM data for DNMT3A2 agree very nicely with the observed methylation pattern, but I am surprised that the DNMT3AC data show the same methylation result (Figs S3C & D). If the construct is the same 300 AA (608-908) one described in Gao et al then it is considerably smaller than the full-length protein, and presumably would show a less perfect alignment with CpG site 57. Are cryo-EM data available for DNMT3AC?
5. Line 358-359: "in DNMT3A2 the PWWP and ADD domains can interact with the H3 tail and help to keep it away from the linker DNA" a reference here would be helpful
6. The data suggest that H3K36me3 stimulates DNMT3AC vs unmodified H3 to a greater extent than it does for DNMT3A2, and it is hypothesized that this is because the catalytic domain lacks the PWWP/ADD domains required to overcome the H3-linker DNA interaction in a nucleosomal context. This would suggest that DNMT3AC should have lower activity vs full-length DNMT3A2 in the presence of unmodified H3. Is this the case (comparing DNMT3A2/DNMT3AC instead of H3K36me3/H3)?

COMMSBIO-21-0570A-Z: “Methylation of recombinant mononucleosomes by DNMT3A demonstrates efficient methylation of linker DNA and a novel role of H3K36me3”

Reply to the reviewers' comments:

Reviewer #1 (Remarks to the Author):

“This paper shows that strong linker-DNA methylation was observed next to the nucleosome core, and the stimulatory effect of H3K36me3 on linker DNA methylation is independent of the DNMT3A-PWWP, where H3K36me3 possibly hindered the H3 tail-linker DNA interaction.

Overall, the data in this paper is well represented and is in good agreement with the previous publications and structures. There are two points that need to be specified or clarified:”

Reply: Thank you very much for this positive assessment.

“1. In Supplemental 2A and 2B, the DNA runs at ~200 bp and nucleosome runs between 500-1000 bp. In Supplemental 2C and 2E, the DNA still runs at ~200 bp, while the nucleosome runs between 2000-3000 bp in the native PAA gel. Is it caused by different gel types, or does the labeling needs to be clarified?”

Reply: Thank you for pointing this out. This difference was due to the different percentages of the gels (6% in S3A, B and 8% in S3E). This point has been clarified in the figure legend.

“2. Since this paper demonstrate a direct stimulatory effect of H3K36me3 on linker DNA methylation independent of the DNMT3A-PWWP domain, and proposed a model of “active gene body-H3K36me3-DNA methylation”, there should be a Figure 3 to represent/explain this mechanism/model. Otherwise, it is too abstract for readers to understand the key points.”

Reply: Thank you very much for this excellent suggestion. Schematic pictures have been included as proposed in the main manuscript (Figure 4E) and Supplement (Supplemental Figure 6E).

Reviewer #2 (Remarks to the Author):

“In this manuscript, Bröhm and colleagues assessed the methyltransferase activity of either full-length or the catalytic domain of Dnmt3a2 against either free or nucleosome-bound DNA. Two observations were made: (i) Nucleosome-bound DNA is protected from methylation by Dnmt3a2 in a manner consistent with the recently published Zjcryo-EM structure; (ii) H3K36 methylation stimulates the Dnmt3a2 activity independent of its PWWP domain. While these findings (especially the H3K36 methylation part) are potentially interesting, the major concern is the physiological relevance of their in vitro assays.

1. It is well-known that adaptor protein either Dnmt3l or Dnmt3b3 is required for Dnmt3a's optimal activity. Indeed, recent paper (PMID: 33004415) shows that re-expression of Dnmt3a2 in the absence

of adaptor proteins has virtually no effect on de novo DNA methylation.”

Reply: We like to mention that in the study cited by the reviewer (PMID: 33004415) low activity of DNMT3A2 in TKO mESCs was observed, but numerous other studies observed active DNA methylation in cell lines after expression of DNMT3A or DNMT3A2 alone (PMID: 29414941; PMID: 32543182; PMID: 33290556; PMID: 25607372; PMID: 32620778; PMID: 30102379). Moreover, our new data also report nucleosomal methylation profiles for the DNMT3A/3B3 heterotetramer, which addresses this issue.

“The cryo-EM structure that was heavily referred to in the study also involved Dnmt3a2-Dnmt3b3 heterotetramer, not Dnmt3a2 alone. Does Dnmt3a2 form homotetramer? It is highly recommended that the authors repeat the assays with either Dnmt3l or Dnmt3b3 added to make their findings physiologically relevant”

Reply: Thank you for this remark. We have now clarified in the manuscript that DNMT3A catalytic domain and DNMT3A2 are forming homotetramers and higher aggregates. This was shown by analytical ultracentrifugation and Size Exclusion Chromatography and the corresponding references are now provided. Please also note that it is not relevant for the interpretation of our data, if DNMT3AC and DNMT3A2 forms a homotetramer or larger oligomers.

We like to point out that the contact region of DNMT3B3 to the acidic patch of the nucleosome is conserved between DNMT3A and DNMT3B suggesting that the DNMT3A tetramer can contact the nucleosome in a similar way as DNMT3B3. This aspect is now clarified in the manuscript and illustrated in new Figure panels showing the alignment (Figure 2C and Supplemental Figure 1). We like to apologize for not providing this important information that justifies our experimental rationale in the previous version of the manuscript and the confusion this might have caused.

Finally, we have now determined the methylation profiles of DNMT3A/3B3 heterotetramers on nucleosomal substrates and added these data to the manuscript. Strikingly, the results show a very similar methylation profile of DNMT3A/3B3 heterotetramers and DNMT3A2 homotetramers, supporting the view that the function of the acidic patch interacting residues is conserved between DNMT3B3 and DNMT3A.

“2. The claim that H3K36me3 stimulates Dnmt3a2 activity independent of PWWP domain is interesting. However, this part needs to be strengthened. The conclusion was made based on comparing full-length vs. the catalytic domain of Dnmt3a2. Dnmt3a2 has other regulatory domains (eg. ADD) in addition to PWWP domain. So the better comparison here should be PWWP-deleted or mutated Dnmt3a2 (in complex with Dnmt3l or Dnmt3b3 as discussed above). Also trimethyllysine analog was used to mimic H3K36me3, the study needs to be verified with true H3K36me3-modified nucleosomes.”

Reply: We understand this question. However, please note, that we observed the K36me3 stimulation in the absence of both domains (PWWP and ADD) indicating that both of them are not involved in the mechanism of stimulation. This has been clarified in the manuscript. Hence additional experiments with a construct including the ADD domain could not add much.

Unfortunately, we do not have the technology to prepare nucleosomes containing methylated K36. We have discussed this caveat in the manuscript now and provided references showing that PWWP domains (including that of DNMT3A) bind the H3K36me3 analog containing peptides. Moreover, we now provide additional fluorescence spectroscopy data directly showing that H3K36me3 reduces the interaction of the H3-tail with the linker DNA, as proposed in our model.

“3. Related to the above point, authors suggest that H3K36me3 may affect DNA-histone interaction, thus allowing better engagement of Dnmt3a2. This suggests that H3K36me3 could act in cis which is a quite provocative idea. Therefore this conclusion needs to be supported by further evidence and structural or biophysical analysis of the H3K36me3 nucleosome.”

Reply: We like to mention that there is emerging recognition of the relevance of the histone tail-linker DNA interaction, which has been documented in our manuscript by references. We have expanded this section and mentioned the experimental approaches leading to this conclusion and added some more references as this aspect was important for the reviewer. Moreover, we have now clarified that a role of H3K36 methylation on this binding is proposed by us on the basis of our data.

While we believe further structural analyses are beyond the scope of the current manuscript, we have conducted an additional biochemical experiment and now show that the presence of H3K36me3 directly affects the fluorescence of an artificially introduced Trp in the H3 tail at position 15 (A15W mutation). This result indicates that the DNA interaction of the H3 tail (and corresponding quenching of the Trp fluorescence) is reduced by H3K36me3, as we also deduced from our kinetic studies.

Reviewer #3 (Remarks to the Author):

“The manuscript followed recent cryo-EM structure of Dnmt3a2-3b3 and performed DNA methylation assays of Dnmt3a2 (full length protein and the catalytic domain) in a nucleosomal context, with and without H3K36 methylation. The authors concluded that (1) the methylation occurs mostly on the linker DNA (where the DNA fragment was not protected by histones), (2) the difference of the DNA methylation level at a given CpG site is largely influenced by neighboring sequence, and (3) H3K36 methylation stimulates DNA methylation by both the full-length enzyme and the catalytic domain, regardless of whether the histone binding domain (PWWP) existed. Given that the preferential methylation of linker DNA has been abundantly documented, the finding that the H3K36me3 stimulates the methylation of linker DNA is of interest but that essentially is the only new finding presented. There are no new mechanistic or unexpected insights and the work is limited by the failure to include an accessory protein in the assays given what is known about the essential function of these proteins.”

Reply: We like to mention that we present the first high-resolution nucleosome methylation data using a linker DNA that is sufficiently long. This allowed us to determine a very characteristic methylation pattern of the sites 57 and 61 for the first time, which directly connects the DNMT3A2-nucleosome structure with kinetic DNA methylation data. By conducting competitive methylation experiments of unmodified and modified nucleosomes, we provide for the first time really quantitative biochemical data on the effects of H3K4me3 and H3K36me3 on DNA methylation by DNMT3A, because this approach compensates for different qualities and concentrations of the nucleosome preparations. Using this approach, we (unexpectedly) showed that the stimulation of

DNA methylation by H3K36me3 is not dependent on the PWWP domain interaction with the H3 tail. In our revised manuscript, we now also show the effect of H3K3 methylation on nucleosome methylation and we present the comparison of methylation data for DNMT3A2 homotetramers and DNMT3A/3B3 heterotetramers, which is also new.

“Below are the detailed concerns with this manuscript: (1) the in vitro experimental setting is different from that of the structural data that the authors used to justify their results, and (2) the conclusions are contradictory to what was previously published from the same laboratory.”

Reply: Please note that we do not use the structural data to justify our results, but we observe that details of the methylation patterns observed with DNMT3A fit very nicely to the structural data. What we rather do is to interpret our biochemical data in context of structure, because of their very convincing overall agreement. Please also note that as described below, there are no contradictions with our previous work.

“In cells, Dnmt3a is regulated by Dnmt3L in ES cells or Dnmt3b3 (or other Dnmt3b inactive isoforms) in somatic cells. In the latest cryo-EM structure, Peter Jones and his colleagues used Dnmt3a2-Dnmt3b3 in complex with nucleosome, where Dnmt3b3 is in contact with nucleosome disk surface. However, in the current study, the authors used Dnmt3a2 alone. There is no biophysical data to support Dnmt3a2 alone forms “tetramer”. As a matter of fact, a paper from the same lab claimed Dnmt3a forms oligomer (reference 13). Thus, the experiment should be performed using the same configuration of Dnmt3a2-Dnmt3b3.”

Reply: Thank you for this comment. We like to point out that the contact region of DNMT3B3 to the acidic patch of the nucleosome is conserved between DNMT3A and DNMT3B suggesting that the DNMT3A tetramer can contact the nucleosome in a similar way as DNMT3B3. This aspect is now clarified in the manuscript and illustrated in new Figure panels showing the alignment (Figure 2C and Supplemental Figure 1). We like to apologize for not providing this important information that justifies our experimental rationale in the previous version of the manuscript and the confusion this might have caused.

We have now clarified that DNMT3A catalytic domain and DNMT3A2 are forming homotetramers and higher aggregates. This was shown by analytical ultracentrifugation and Size Exclusion Chromatography and the corresponding references are now provided. Please also note, that it is not relevant for the interpretation of our data, if DNMT3AC or DNMT3A2 form homotetramers or larger oligomers.

Finally, we have now determined the methylation profiles of DNMT3A/3B3 heterotetramers on nucleosomal substrates and added these data to the manuscript. Strikingly, the results show a very similar methylation profile of DNMT3A/3B3 heterotetramers and DNMT3A2 homotetramers, supporting the view that the function of the acidic patch interacting residues is conserved between DNMT3B3 and DNMT3A.

“The main question the authors addressed in the manuscript has been addressed by the same laboratory previously (reference 15). In 2005, they used Dnmt3a full length and the catalytic domain, same as used in the current study, yet the opposite conclusions were made (see below). Although the

technique used to detect methylation is different, the principle is the same (bisulfite sequencing), and the same enzymes were used. The differences should be fully addressed. A solid conclusion should not change as a function of time."

Reply: Please note, that there is a key difference in the design of the older experiment, which is that we used nucleosomes only based on the MMTV sequence with 146 bps without extended linker DNA regions, which (as we know now) are the main sites of DNA methylation. Therefore, unfortunately, the design of the previous study was not ideal from the current perspective, but please note that more than 15 years passed since the inception of this work and our current level of understanding. Due to the omission of linker DNA, the previous study was lacking an essential structural feature not only needed as main methylation target, but also to support the interaction of DNMT3A with the nucleosome and histone tails. Therefore, the interaction of the DNMT with the nucleosome was totally different in the previous work. As a consequence, methylation levels observed by bisulfite were very low. While we agree that the old study from today's view was not ideally designed, we like to point out that its results are still valid, just they cannot be compared with the current experimental setup.

"Examples of different conclusions made in 2005 (reference 15) and now (current manuscript): (2005) The methylation sites are not clustered towards the ends of the DNA, which are typically more accessible than the core region of the nucleosome.

(Current) Strong methylation was observed in the linker DNA region, whereas little methylation activity could be observed for the CpGs within the DNA region that is bound to the nucleosomes."

Reply: In the previous paper, there was no linker DNAs included. Hence methylation could not have occurred in the linker region.

"(2005) No significant difference in binding and methylation between all four tail-less histone variants. No difference in the activity of Dnmt3a on H3K9methylated nucleosome.

(Current) Nucleosome containing H3K36methylation analogs stimulates Dnmt3a activity, even under the context of without histone tail binding domain (i.e., Dnmt3a catalytic domain alone)."

Reply: One key result of our new work is that the interaction of the H3 tail with the linker DNA regulated the activity of DNMT3A. This observation could not have been made in the previous study lacking the linker DNA.

"Is the stimulation effect specific for H3K36 methylation? How about H3K4me3 and H3K9me3 in the absence of histone binding?"

Reply: We agree with the reviewer that this is an important and interesting question. As there is no reported molecular interaction between DNMT3A and H3K9me3, we have focused on H3K4me3 and conducted another set of experiments with H3K4me3 containing nucleosomes. Our data show that the reduced interaction of DNMT3A2 with the H3K4me3 containing H3-tails leads to a specific

reduction of methylation of the CpG sites close to the nucleosome core. Hence while H3K_c36me₃ increased DNA methylation H3K_c4me₃ did the opposite.

“(2005) CGG is the preferred site of methylation.

(Current) Figure 1D, please show the sequence information.

Reference 10 showed Dnmt3a has a different preference (TnCG-c/t-c/t).”

Reply: Please note that the sequence of all sites is specified in Figure 1A. The numbering of the sites is provided in Fig. 1D. As mentioned above, the 2005 paper observed only few methylation events, which all occurred on the core histone DNA which required complex interaction and likely stripping of the DNA from the nucleosome. The (weak) methylation preferences for these sites in a nucleosomal context cannot be compared with the methylation of free linker DNA or unbound DNA as studied in ref. 10.

“(2005) Methylation events at non-CpG sites.

(Current) no mention of non-CpG methylation”

Reply: We did not collect information about non-CpG methylation in this work.

“(2005) methylation activity of Dnmt3 catalytic domain was strongly inhibited by nucleosomal DNA.

Binding of Dnmt3a catalytic domain to nucleosome was very weak.

(Current) Do these observations still hold?”

Reply: This observation is absolutely valid. Methylation of DNA within the nucleosome region was strongly reduced with DNMT3A2 and DNMT3A catalytic domain as shown in Figure 1C, 3A, 4B, and 4C, as well as Suppl. Fig. 4. The second question refers to binding studies, which we did not conduct in the current investigation.

Reviewer #4 (Remarks to the Author):

“This study investigates the impact of trimethylation of histone H3 at lysine 36 (H3K36me₃) on DNA methylation carried out by the methyltransferase DNMT3A2. Using an in vitro methylation assay on reconstituted nucleosomes containing DNA based on the Widom-601 sequence with an extended 5' linker, the authors demonstrate that DNMT3A2 shows a preference for certain CpG sites in a nucleosomal context vs free DNA, and the presence of a histone octamer protects against methylation of octamer-bound DNA. Furthermore, H3K36me₃-containing nucleosomes correlate with higher levels of methylation when compared to those containing unmodified H3, and this effect is shown to be more dramatic when the assay is performed using the DNMT3A catalytic domain (DNMT3AC).

Finally, using cryo-EM data it is demonstrated that the main site preferentially methylated by DNMT3A2 in the presence of a nucleosome (site 57) is situated close to the catalytic core of one of the DNMT3A2 subunits in the

heterotetramer formed by DNMT3A/DNMT3B3 when bound to the nucleosome.

This manuscript is well-written, and the methods used are nicely described and easy to follow. The

claims made are for the most part well-supported by the data, and a compelling model is presented for the role and evolution of the relationship between DNA methylation and H3K36me3. There are some (mostly minor) comments/questions outlined below. I would consider points #4 & #6 most important, as they potentially impact the model presented in this article.”

Reply: Thank you very much for this positive assessment.

“Comments/questions:

1. Fig. S1A: perhaps label black line as “H3Kc36me3” instead of “H3K36me3” to make it obvious that this is the converted sample rather than some other H3K36me3 peptide”

Reply: Thank you. This has been changed as proposed.

“2. Table S1: DNMT3A2 WT H3Kc36me3 repeats #1 & 2 have far fewer reads than the H3 samples. Do you know why this might be the case?”

Reply: These were initially studied where some steps in the processing of the libraries were not yet optimized. Still the results had enough reads to be meaningful, this is why we included them in the analysis. Omission of these data does not change the results (see figure R1).

All replicates

w/o two experiments with low read numbers

Figure R1: Comparison of the analysis of the H3 and H3K36me3 methylation data with DNMT3A2 either using the full data set or after excluding the two experiments with lower read numbers.

“3. Fig. 1C. What order were the 2 free DNA replicates done relative to the nucleosomal replicates? Two of the nucleosomal replicates at site 57 are quite close to the free DNA replicates. Were these done together and the subsequent (higher) nucleosomal replicates done before/after?”

Reply: Please note, that the experiments with the different nucleosomes and free DNA were not connected, so it does not make sense to compare their order. Direct comparison is only possible for the experimental sets in which unmodified and H3K₃₆me₃ or unmodified and H3K₄me₃ modified nucleosomes were methylated in competition. The paired data were always analyzed together in our study.

“4. The cryo-EM data for DNMT3A2 agree very nicely with the observed methylation pattern, but I am surprised that the DNMT3AC data show the same methylation result (Figs S3C & D). If the construct is the same 300 AA (608-908) one described in Gao et al then it is considerably smaller than the full-length protein, and presumably would show a less perfect alignment with CpG site 57. Are cryo-EM data available for DNMT3AC?”

Reply: Please note that in the cryo-EM picture only the catalytic domain of DNMT3A and C-terminal part of DNMT3L are resolved. This has now been indicated in the new Figure panel 2C and Suppl. Figure 1. Hence all visible structural elements are present in DNMT3AC and DNMT3B3C, including the very nice positioning of site 57 just below one active center.

“5. Line 358-359: “in DNMT3A2 the PWWP and ADD domains can interact with the H3 tail and help to keep it away from the linker DNA” a reference here would be helpful.”

Reply: Please note that this sentence describes our mechanistic model. This point has been clarified in the manuscript.

“6. The data suggest that H3K₃₆me₃ stimulates DNMT3AC vs unmodified H3 to a greater extent than it does for DNMT3A2, and it is hypothesized that this is because the catalytic domain lacks the PWWP/ADD domains required to overcome the H3-linker DNA interaction in a nucleosomal context. This would suggest that DNMT3AC should have lower activity vs full-length DNMT3A2 in the presence of unmodified H3. Is this the case (comparing DNMT3A2/DNMT3AC instead of H3K₃₆me₃/H3)?”

Reply: This is a very interesting question. However, things are more complicated, because the ADD domain of DNMT3A has an autoinhibitory role as well. Therefore, DNMT3A catalytic domain has a higher activity towards naked DNA (PMID: 25383530; PMID: 20223770).

REVIEWERS' COMMENTS:

Reviewer #1 (Remarks to the Author):

This paper shows that strong linker-DNA methylation was observed next to the nucleosome core, while the nucleosome-bound DNA is protected from methylation by DNMT3A2. In addition, the stimulatory effect of H3K36me3 on linker DNA methylation is independent of the DNMT3A-PWWP, where H3K36me3 possibly hindered the H3 tail-linker DNA interaction.

First of all, the two points that needed to be specified or clarified are clear now:

1. In Supplemental 2A and 2B, the DNA runs at ~200 bp and nucleosome runs between 500-1000 bp. In Supplemental 2C and 2E, the DNA still runs at ~200 bp, while the nucleosome runs between 2000-3000 bp in the native PAA gel. Is it caused by different gel types, or does the labeling needs to be clarified? Clarified.

2. Since this paper demonstrate a direct stimulatory effect of H3K36me3 on linker DNA methylation independent of the DNMT3A-PWWP domain, and proposed a model of "active gene body-H3K36me3-DNA methylation", there should be a Figure 3 to represent/explain this mechanism/model. Otherwise, it is too abstract for readers to understand the key points. Added as Figure 4E.

Second, more work is now done and addressed in the manuscript, such as the effects and comparison of methylation between H3K4me3 and H3K36me3, thereby presenting more new findings compared to the initial version.

Third, since the methylation effects of H3K4me3 was mentioned, it would be better to include H3K4me3 mechanistic hints in the Figure 4E schematics.

In principle, the data now in this paper is well represented and is in good agreement with the previous publications and especially the Cryo EM structures.

Reviewer #2 (Remarks to the Author):

The authors have satisfactorily addressed my concerns and I recommend acceptance. This work represents a significant advance in our understanding of de novo DNA methylation.

Reviewer #3 (Remarks to the Author):

I have no further comments.

Reviewer #4 (Remarks to the Author):

I am satisfied that the authors have addressed the points that I raised in the initial review